# The Potential Roles of the N and P Supplies on the Internal Browning Incidence in Sweet Cherries in the Southern Chile

Cristóbal Palacios-Peralta [1], Marjorie Reyes-Díaz [2,3,*] , Jorge González-Villagra [4,5] and Alejandra Ribera-Fonseca [1,3,*]

1 Centro de Fruticultura, Facultad de Ciencias Agropecuarias y Forestales, Campus Andrés Bello, Universidad de La Frontera, Avenida Francisco Salazar 01145, Temuco P.O. Box 54-D, Chile

2 Departamento de Ciencias Químicas y Recursos Naturales, Facultad de Ingeniería y Ciencias, Campus Andrés Bello, Universidad de La Frontera, Avenida Francisco Salazar 01145, Temuco P.O. Box 54-D, Chile

3 Center of Plant-Soil Interaction and Natural Resources Biotechnology, Scientific and Technological, Bioresource Nucleus (BIOREN), Campus Andrés Bello, Universidad de La Frontera, Avenida Francisco Salazar 01145, Temuco P.O. Box 54-D, Chile

4 Departamento de Ciencias Agropecuarias y Acuícolas, Facultad de Recursos Naturales, Universidad Católica de Temuco, Temuco P.O. Box 15-D, Chile

5 Núcleo de Investigación en Producción Alimentaria, Facultad de Recursos Naturales, Universidad Católica de Temuco, Temuco P.O. Box 15-D, Chile

* Correspondence: marjorie.reyes@ufrontera.cl (M.R.-D.); alejandra.ribera@ufrontera.cl (A.R.-F.)

**Abstract:** Southern Chile has experienced a strong increase in sweet cherry production in recent years. However, there are climatic gaps that negatively reduce the fruit quality and yield of exportable fruit, such as the high incidence of rains during flowering and pre-harvest. The use of roof covers has become an agronomic solution that offers protection from weather events, such as rain, which will significantly increase the exportable fruit. However, the use of plastic covers can cause negative impacts on the fruit, such as a loss of firmness and acidity. Currently, the incidence of internal (pulp) browning has reduced the quality of cherries produced in Southern Chile, although research on this subject is largely under explored. Some studies reported that a high content of antioxidants in the fruit, both phenolic and non-phenolic (e.g., ascorbic acid), could reduce the incidence of the physiological disorder of browning. The soils of Southern Chile are characterized by the high content of organic matter, which implies high levels of available nitrogen (N) and a high phosphorus (P) content. Some studies, however, have shown that fertilization with N and P would significantly improve the postharvest quality, shelf life, and the accumulation of antioxidant compounds in fruits, even better than other strategies, including selenium and chitosan applications. However, there needs to be more detailed information on this aspect of the sweet cherry fruit production. The quality attributes and postharvest life of cherries are closely associated with the antioxidant levels of fruits, which could be related to either the soil acidity level of the Chilean Andisols or the levels of P and N in soil or plant tissues. Therefore, the objective of this review was to discuss the role of the N and P supply on the internal browning incidence in sweet cherries and relate it to what is known in other fruits.

**Keywords:** antioxidant composition; inorganic fertilizer; physiological disorders; postharvest quality





## 1. Introduction

Currently, Chile has reached the third place among countries with the highest sweet cherry production worldwide, after Turkey and the United States of America [1,2]. Noteworthy, Chile has shown a strong increase in the sweet cherry production (from 41,000 to 234,000 tons) over the last 10 years. According to a recent report by the USDA (2021), sweet cherry planted areas in Chile reached a total of 49,000 hectares (producing around 395,000 tons) during the 2021/22 season, and these orchards are mainly located in Central Chile (General Bernardo O'Higgins and Maule Regions) [3]. In the last few years, the

plantation area has been expanding into Southern Chile (from La Araucanía to the Los Lagos Regions) [3,4], which is considered an attractive zone for the production of this fruit crop. This expansion is due in part to the chance to cultivate late-harvest cultivars, whose fruits can be sold at high prices during the Chinese New Year [5].

Southern Chile (37°35′–40°33′ S), however, has climatic constraints for the sweet cherry cultivation, mainly the high precipitation levels at bloom and pre-harvest, but also the late spring frost, which reduces the yield [6], the fruit quality, and the physiological condition of the fruit. In this zone, the sweet cherry is mostly cultivated on volcanic ash-derived soils (Andisols), which are acidic (pH ≤ 5.5) [7–10]. In this regard, it is important to mention that the natural soil N supply in volcanic soils, which are rich on organic matter (OM; up to 20%) [11], are a significant source of plant-available N for crops [12]. Moreover, the high contents of OM and allophane in these soils [11] results in a low P availability [13–15]. Annual rainfalls range from 1200 to 2800 mm per year, in this zone [16] resulting in rain-induced fruit cracking as one of the main limiting factors for the sweet cherry production [17,18]. Thus, the use of fruit-cracking resistant cultivars (such as Regina) and the use of plastic covers to protect orchards from rain (mainly at harvest time), are among the most important sweet cherry management methods available to the farmers in Southern Chile [17,19]. Although the use of plastic covers has been reported to be an effective strategy to reduce the rain-induced cracking in sweet cherries, several studies have shown that such covers can also negatively alter the fruit firmness, decreasing the resistance of the fruits to mechanical damage that can occur during harvest, handling, and a long post-harvest trip [20–22]. Moreover, the fruit skin color and acidity can be decreased by covering [17,23,24]. This is a concern because the quality of sweet cherries in the fresh market is usually determined by fruit size, skin color, sweetness, sourness, and firmness [25,26], and also by their nutritional composition, including anthocyanins, quercetin, hydroxycinnamates, potassium, fiber, vitamin C, carotenoids and melatonin which have shown potential preventive health benefits for humans [27,28].

Sweet cherries have one of the higher respiration rates among the non-climacteric fruit species, thus they are susceptible to the rapid senescence after harvest which decreases their postharvest life [17,29,30]. According to Alique et al. [31], these fruits are highly perishable, and their shelf-life is relatively short, due to their significant rates of respiratory activity and susceptibility to rapid microbiological decay during storage. Surface pitting, skin darkening, pedicel browning, flesh softening, loss of flavor, and fungal decay are common deteriorations of the sweet cherry fruit during postharvest [32]. Moreover, internal browning (IB) is recognized as another important defect at postharvest for sweet cherries [18,32,33]. The incidence of IB of Chilean sweet cherries, coincident with herbaceous flavor, increased in the last few years due to the increased shipping time to export destinations, reaching levels ranging from 13% to 42% of incidence (for cv. Regina) when fruits are cold stored for more than 45 days [33] (Figure 1). Indeed, according to reports by the technical committee on the sweet cherry [2], sweet cherries produced in Southern Chile were exhibiting an increase in IB incidence (to levels up to 15%), due to delays in the clearance from customs provoked by difficulties of the global logistics of maritime transport and slowed processes, due to COVID problems. The increments of trip time to export destination is an important gap that will continue for the foreseeable future, caused by the growth of fruit volume of the late-harvested varieties (such as Regina) arriving at the same, to the Chinese market; the main export destination of Chilean sweet cherries [34].

It is noteworthy that the committee concurs with the early research that indicated that fruits with a low firmness and low acidity seem to be most susceptible to IB. Nonetheless, this hypothesis still needs to be further tested. Furthermore, it should be noted that almost all studies regarding IB in sweet cherries have focused on the impact of postharvest storage conditions [34–37]. Thus, research about the influence of the pre-harvest factors on the IB modulation is a key challenge to fully understand the problem in the sweet cherries produced in Southern Chile. In this way, it has been shown that some preharvest strategies to avoid this disorder, such as selenium spraying in pre harvest, demonstrated a positive effect on the parameters of fruit composition, internal browning and fruit quality

at post-harvest through a decrease in the polyphenol oxidase activity and an increase in the antioxidant composition in fruits [38–41]. Nevertheless, nutritional factors, such as the appropriate fertilization doses of calcium (Ca), nitrogen (N), and phosphorus (P), can reduce IB in fresh fruits of the Rosaceae, such as peaches [42,43]. Therefore, the objective of this review was to discuss what has been reported on the role of the N and P supplies on the internal browning incidence in sweet cherries, due to the relevant conditions of the soil in Southern Chile and to critically examine these results, in relation to what is known for other fruits, primarily the Rosaceae.

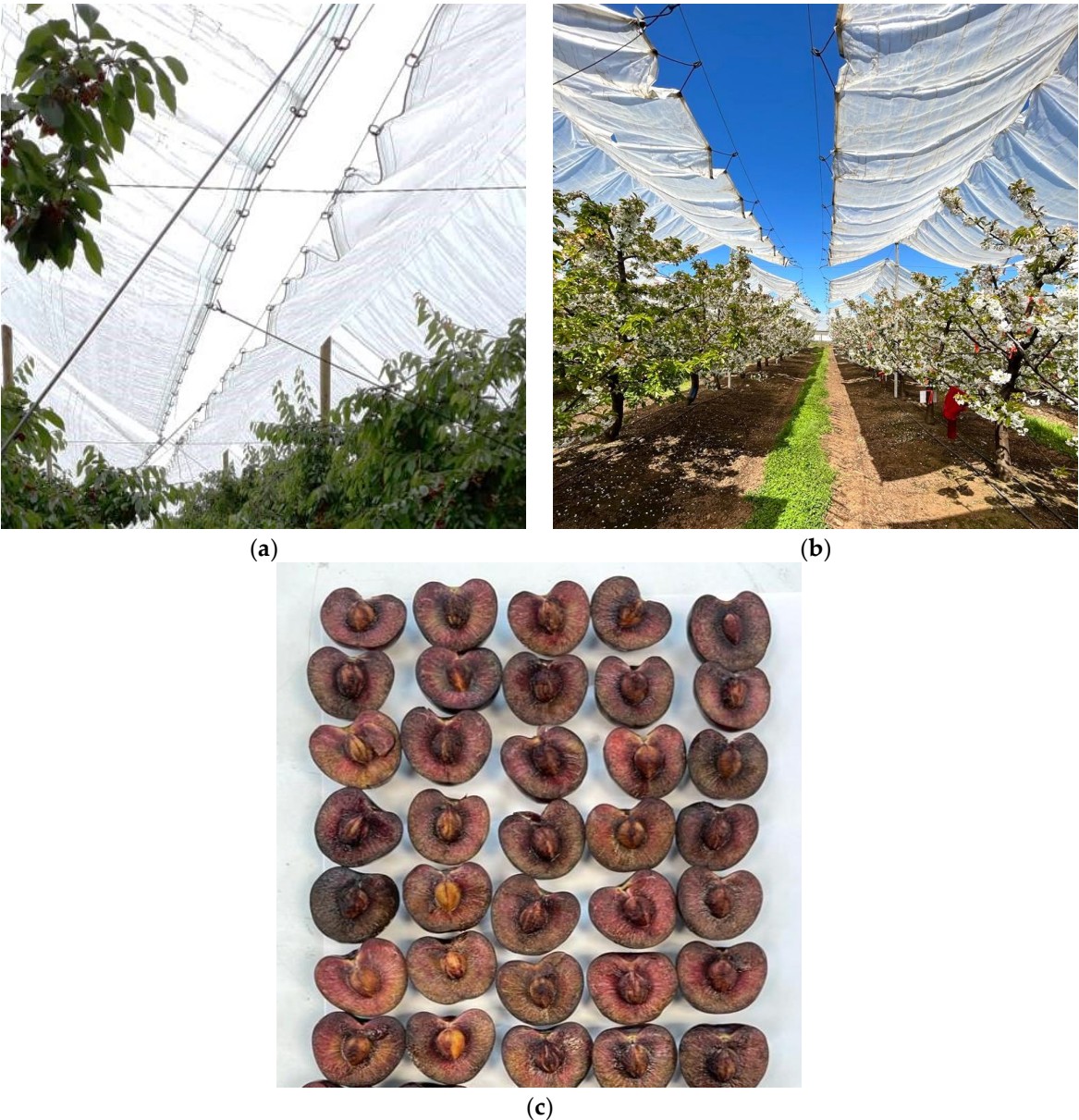

**Figure 1.** Plastic cover used in orchards in Southern Chile (**a**), Images of a sweet cherry orchard in South-ern Chile (**b**) and the sweet cherry internal browning at postharvest (**c**).

## 2. Andisols of Southern Chile: A Particular Scenario for the Sweet Cherry Production

The soil acidity of Andisols should be considered a primary risk factor for the sweet cherry production in Southern Chile. In this zone, the sweet cherry is mostly cultivated on volcanic ash-derived soils (Andisols), which are acidic (pH ≤ 5.5) and have high amounts of available aluminum and manganese, resulting in the well-known negative impacts on plant growth, as well as on the yield and quality of fruit crops [8–10,13]. The high acidity

of these soils is mainly derived from the high rainfall in this zone (mainly in winter), which results in the leaching of metal cations, and this is heightened from the use of the urea anion as the main N fertilizer [44,45]. The rich OM in these volcanic soils also forms a natural soil N supply, (up to 20%) [11], and this is a significant source of plant-available N for crops. However, intensive farming systems frequently still consider supplementation with the application of high fertilization rates, even to such soils in Southern Chile [44]. Moreover, the high contents of OM and aluminum silicate clay allophane in these soils [11] also results in low the P availability to crops, due to the increased P fixed on the reactive soil surfaces [13–15]. Indeed, plant P available commonly ranges from 7 to 15 mg kg$^{-1}$ soil in the Andisols of Southern Chile [44]. In this regard, during the last few decades, the amendment applications, such as lime, dolomite, or gypsum, have been an agronomic practice commonly used to decrease the phosphate sorption on Andisols, ameliorating the negative impacts of the soil acidity on the plant production [13,44].

The sweet cherry can grow successfully in soils with OM contents ranging from <1% to >10%. The recommended soil pH is within the range from 6.0 to 7.0, while with pH values as low as 5.5, it can be seen to inhibit plant growth [46,47]. Interestingly, studies have shown that many of the deleterious effects of the acidity complex of Andisols on plant growth can be overcome by increasing the P and N supplies [48]. It is important to point out that the mineral requirement of the sweet cherry is medium to high for N and relatively low for P. Nevertheless, when the soil pH is low, the P supply becomes a key factor in order to ensure a good vigor and yield levels [46,47,49]. According to Hirzel [50], the appropriate N and P soil availability for the sweet cherry production ranged from 15 to 40 mg kg$^{-1}$ and 15 to 20 mg kg$^{-1}$ soil, respectively. Despite these antecedent issues, there needs to be more information about the N and P requirements of the sweet cherry grown in the Andisols of Southern Chile under plastic covers, and more knowledge about how the macronutrients can modulate the fruit quality and condition at harvest and postharvest.

## 3. Physiological and Biochemical Traits of the Internal Browning in Fresh Fruits

Fruit internal browning (IB) is a consequence of the oxidative degradation of the phenolics by the polyphenol oxidase (PPOs) enzymes, which leads to the production of quinones that, in turn, rapidly polymerize to form the characteristic brown-colored products [51–54]. The IB disorder is a major industrial concern due to the negative impacts of the sensory and visual traits and quality loss of the fruits [42]. It can be triggered by maturation; fruit bruising; exposure to oxygen in fresh cut, sliced, and pulped forms; or by thawing fruit after prolonged freezing [42,55]. The initial enzymatic oxidation occurs when the phenolics stored in the vacuole are acted upon by PPOs located in the cytoplasm, to form slightly colored quinones, generating a deteriorative process for the fruits [53]. The incidence of IB in plant tissues depends on the activity of the PPO, the presence of available oxygen, and the concentration of the phenolic compounds [56,57]. In sweet cherries, the activity of the PPO increases during ripening and storage, for example after 50 days at −1 °C, a higher PPO activity is found showing a parallel tendency with the observed pectin methyl esterase activity [58,59]. Interestingly, many phenolics, including caffeic esters and catechins, act as good antioxidants at relatively low concentrations, while, at higher levels, they are susceptible to oxidation themselves (they behave as pro-oxidants) [53] and treatment with phenolics can be helpful in limiting the postharvest decay in the sweet cherry [60]. Moreover, it has been reported that IB showed a significant positive correlation with the chlorogenic acid content in peach and nectarine extracts [61]. According to Jiang et al. [62], the chlorogenic acid oxidation proceeded rapidly and is also implicated as an important factor in IB of coconut fruit, as well as the acid levels. Chlorogenic acid and (ÿ)-epicatechin are the two major compounds found in pears and apples, acting as endogenous substrates for PPOs [63]. Similarly, flavanols were degraded during oxidation, contributing largely to the brown pigment development in pears, while some oligomers (procyanidins) have also been implicated in the IB of pears [53]. Lea [64] reported that flavonoids are relatively slowly attacked by the oxidative enzymes, such as o-phenoloxidases. In the presence of transfer substances, such as catechins and chlorogenic acids, they are oxidized more rapidly with the probable formation of dimers, as

a first step [65]. Ju et al. [66] concluded that simple phenols initiated the oxidation process in apples, but the flavonoids produced brown colors by polymerization, acting as protection for the anthocyanins. In addition to antioxidant phenols, a high concentration of ascorbic acid is also highly desirable in fruits during storage, since it may prevent the enzymatic browning by inhibiting the PPO activity [67]. This is because quinones can be reduced by the antioxidant activity (AA), regenerating phenolics, and inhibiting the fruit IB [42]. Thus, ascorbic acid could be an excellent alternative to increase antioxidants and thus to decrease IB in fruits.

## 4. Non-Enzymatic Antioxidant Compounds in Sweet Cherry Fruits

The connection between oxidative damage and senescence in plant species has been widely recognized [68]. The theory proposed by Szilard [69] and Orgel [70] relates the process of senescence with the accumulative molecular damage caused by the reactive oxygen species (ROS) [71]. Thus, a decline in the fruit postharvest quality can be primarily associated with decay and senescence induced by oxidative stress [68]. The increased membrane permeability is largely explained by lipid peroxidation catalyzed by the free radicals in plant senescent tissues [72,73]. Thus, an enhanced pool of antioxidants in fruits may extend their shelf-life and improve the postharvest quality by delaying senescence [66,72]; reviewed in [73]. Noteworthy, sweet cherries are highly valued by consumers, not only due to their taste but also their nutritional value and the health effects related to their high levels of fiber, carotenoids, phenolic compounds, and ascorbic acid [17,74–76].

Phenolic compounds are the major phytochemical group in sweet cherries, including phenolic acids, tartaric esters, flavonols, and anthocyanins [27,77,78]. All of these molecules contribute to the total fruit antioxidant activity [79–81], playing a protective role against the oxidation processes occurring on fruits, mainly during postharvest [29,82]. Anthocyanins have been reported as the main phenolic compounds in sweet cherries, as the aglycon cyanidin bound to the glycosides 3-rutinoside and 3-glucoside, which are the most important [77,83]. The phenolic compounds in these fruits also include flavan-3-ols, flavonols, and non-flavonoids, such as hydroxycinnamic and hydroxybenzoic acids [81,84]. Interestingly, Commisso et al. [85] reported that the composition of sweet cherry cultivars depended more on the genetic variability than on environmental factors, with phenolic compounds being the principal source of antioxidant activity in these fruits. Furthermore, they found a strong synergy between the anthocyanins and quercetins/ascorbic acid, specifically in some cultivars, and concluded that the total antioxidant activity of sweet cherry fruits may originate from cultivar-dependent interactions among different classes of metabolites.

In addition to phenolic compounds, ascorbic acid is one of the major secondary metabolites produced in sweet cherry fruits [86]. The ascorbate-glutathione cycle, which maintains the AA levels in plant tissues by controlling the biosynthesis, oxidation, and recycling of antioxidants, participates in the detoxification of the ROS and thus promotes the plant tissue resistance to senescence and environmental stresses [87,88]. The ascorbic acid (vitamin C) content largely varies between plant species, The value in sweet cherries is around 7 mg $100 \text{ g}^{-1}$ FW, whereas citrus fruits reach up to 75 mg $100 \text{ g}^{-1}$ FW [89,90]. In addition to the direct action of these antioxidant compounds, several studies have shown that the bioactive composition of fresh fruits can also be affected by the different pre-harvest factors, including the mineral supply [90–92].

## 5. Nitrogen Excess Could Reduce the Quality, Condition, and Antioxidant Levels in Fruits

An adequate soil supply of nitrogen (N), one of the critical limiting factors for the growth of plants, is also essential for quality fruit crops [92,93], and adequate N allows for the good development of fruit color, flavor, texture, and nutritional quality of fruits [94]. Moreover, studies have shown that a N deficiency in the leaf may result in color alterations and reduce the fruit size [95], while an excess of N has been associated with smaller and deformed fruits [96–98]. Furthermore, N fertilization has been shown to decrease the firmness in many fruits [99–101]. The total marketable melon fruit yield and fruit N content showed a concomitant increase

in response to the N levels applied [102]. Cheng et al. [103] found that when the N supply was increased in apples, the concentrations of sugars (sucrose, glucose, fructose) and the total nonstructural carbohydrates decreased, increasing the concentration of the organic acids. The uptake and assimilation of N by the plants can be influenced by both the source and the doses of N applied [104,105]. Plants acquire inorganic N from the soil, mainly in the ammonium ($NH_4^+$) and nitrate ($NO_3^-$) forms [106]. When $NH_4^+$ is only applied as a unique N source, it increases the internal ammonia ($NH_3$) levels, which can be highly toxic for the plant cells [107,108]. Moreover, it has been shown that the Ca, magnesium (Mg), and K uptake is usually higher for $NO_3$-N-grown plants [109]. Mengel and Kirkby [107] reported that tomato plants supplied with $NH_4^+$ as a unique source of N, tended to have a higher incidence of blossom-end rot caused by a Ca shortage. A Ca reduction in fruit tissue induced by high $NH_4^+$ may reduce the harvest quality and particularly the postharvest life of fruits [109]. High N fertilization (15 M m$^{-3}$) has been associated with a shorter postharvest life of some fruits, due to the increased susceptibility to mechanical damage, physiological disorders, and decay [108]. Indeed, the N supply can affect the postharvest life of many perishable fruits, including strawberries and sweet cherries [102,110–112]. A high N content in fruits was also related to high breakdown incidences of apple fruits during storage [113]. Smock and Gross [114] reported the stimulation of the respiratory activity in apple fruits, whereas a classic study be Gourley and Hopkins [115] did not detect any respiration change in response to the N supply.

It has been reported that the availability of inorganic N influences the synthesis of secondary metabolites in plants; however, the results from multiple studies appear contradictory. Nonetheless, the lack of N might trigger the ROS accumulation and the concomitant oxidative damage, due to the alterations in photosynthetic performance [116,117]. Nevertheless, the activity of the primary antioxidant enzyme, superoxide dismutase [118], and phenolic compounds, such as anthocyanins [119], might counteract the oxidative stress under N starvation. The precise application of N fertilizer should positively influence the fruit quality, including the anthocyanin content [120]. In contrast, many studies suggest that with the increasing N supply, a decrease in the anthocyanin synthesis is observed while it concomitantly increases the chlorophyll content in fruits. Under the highest treatment of N nutrition, the shoot growth was enhanced in apple trees, whereas the activity of phenylalanine ammonia lyase (PAL; a key enzyme involved in the phenols biosynthesis) seemed to be down-regulated, resulting in a generally decreased flavonoid accumulation [121] (Figure 2). According to Wen et al. [122], high N leads to a higher chlorophyll concentration in the fruit skin, thus increasing the fruit greenness, while the nitrate deficiency suppresses the N absorption and assimilation, in turn suppressing the chlorophyll synthesis in apple trees. In agreement, Rubio Ames et al. [123] found that peach fruits with higher N rates were greener and had higher total soluble solids than fruit from the trees grown under low N.

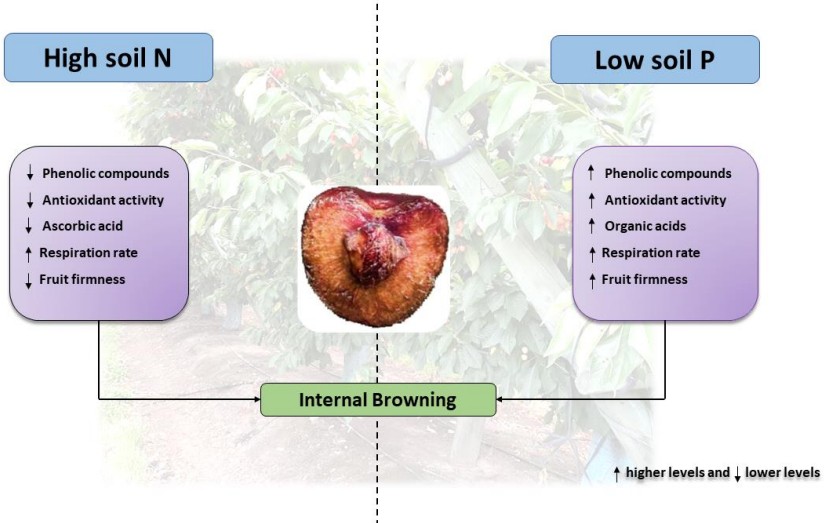

**Figure 2.** Influence of N and P fertilization on the internal browning in the sweet cherry.

Interestingly, it has been shown that the application of the nitrification inhibitor, 3,4-dimethyl pyrazole phosphate (DMPP), overcomes the problem of poor fruit coloring caused by the excessive N supply in apple trees [124,125]. This is likely because the DMPP reduced the capacity of the N absorption and accumulation in fruits and whole apple plants, whereas it increased fruit anthocyanin and solid soluble content (SSC) without effects on the yield [125]. Studies have demonstrated an increase of polyphenols in grape berries when increasing amounts of N fertilizer were applied [126]. Likewise, when strawberries were treated with three levels of nitrate $NO_3^-$, the highest dose (15 M m$^{-3}$) produced the maximum fruit yield. However, the nutraceutical quality, such as the antioxidant capacity and phenolic compounds, significantly declined [106]. Under a N deficiency, the increased phenolic compounds and antioxidant activity have also been reported in mustard, sweet basil, and lettuce [126–128]. These findings agree with the Bryant et al. [129] carbon/nutrient balance (CNB) hypothesis, which sustains that plants growing under poor N have more secondary metabolites than those in a N-rich environment. In this way, the decrease in the antioxidant contents, including anthocyanins, in response to high N doses seems to be due to the rapid plant growth and development, and the preferential allocation of resources is directed to the growth processes rather than the secondary metabolism [91,92,130,131]. One concept is that the N excess would be diverted into amino acids and protein synthesis, whereas the N-deprived plants would allocate C resources towards the phenolic compound production [129,132]. Consistent with this idea is the report that there is a depletion of C compounds, that serve as substrates for the amino acid production in Arabidopsis plants grown under an excessive N supply [133]. According to Stewart et al. [134], the antioxidant compounds are induced in the N-starved plants as a consequence of the increased ROS levels, while plants growing with a high N supply decrease their concentration of phenolic compounds. Based on this analysis, the decrease in the N supply for improving the antioxidant capacity could be used as a strategic tool to enhance both the quality and the profitability of the crops, minimizing the environmental impact of N [135]. In fact, our previous studies indicated that the levels of the total phenolic and total anthocyanin in leaves of the highbush blueberry cultivars were reduced in response to the increasing N supply [136]. Moreover, we found that it appears that the leaf N concentration over 15 g N kg$^{-1}$ could represent a critical threshold of the N level above which the phenols (mainly anthocyanins) decrease in blueberry leaves [136].

It has been reported that N applications reduced the vitamin C content in blackberries [137], cauliflower [138], juices derived from oranges, lemons, and grapefruits [139], white cabbage [140,141], and crisp-head lettuce [140] (Figure 2). Cardeñosa et al. [142] concluded that applying high N amounts could lead to an increase in the citric acid content and thus conceal the sweet fruit flavor in strawberries, whereas low N levels could decrease the acidity and result in an observably lower content of AA and a higher fruit firmness. The application of the low level of N by fertigation, however, had no deleterious effect on the fruit of the sweet cherry (Table 1) [112], while higher N showed deleterious effects in the sour and Chinese cherry [143–145]. Furthermore, Augustin [146] reported a decrease in AA of some cultivars of potatoes, with increasing amounts of N fertilizer used. Lisiewska and Kmiecik [138] reported that increasing the amount of N fertilizer from 80 to 120 kg ha$^{-1}$, decreased the vitamin C content by 7% in cauliflower. Additionally, N fertilizers are also known to increase the plant foliage and thus may reduce the light intensity and accumulation of AA in the shaded parts [147].

**Table 1.** Studies about the impact of nitrogen fertilization on antioxidants and the quality in fruits in different plant species.

| Species | Treatments | Nitrogen Treatment | Effects | References |
|---|---|---|---|---|
| *Prunus avium* L. | Fertigation | Annual Rate 160 kg ha$^{-1}$ Decreasing monthly application | No effect with last month N | [112] |
| *Prunus cerasus* L. | 34% ammonium nitrate | 0, 60, 120 kg N ha$^{-1}$ | Reduction of fruit weight, extract content, acidity, and ratio of the total soluble solids to a titratable acidity | [143,144] |
| *Prunus pseudocerasus Lindl.* | Urea | 0–0.8 kg plot | N content showed an optimum below the highest dose | [145] |
| *Malus domestica* L. | 0–5–10–15–20 mM NH$_4$NO$_3$ | 15–20 mM NH$_4$NO$_3$ | Increase nonstructural carbohydrates increase the nitrogen concentration in plants, decrease of the C/N ratios | [103] |
| *Prunus Avium* L. | 50% of shading and NO3:NH4 ratios (0:100, 75:25, 50:50 and 25:75) | 75:25 and 50:50—Higher NO$_3$ ratios—Higher NH$_4$ ratios | Increased yield, fruit size. Increased titratable acidity. Increased total soluble content | [109] |
| *Fragaria ananassa* | 0–60–120 Kg N ha$^{-1}$ | 60–120 Kg ha$^{-1}$ | No effect on the fruit quality | [110] |
| *Brassica Juncea* Coss | 10–25 mM N and 0.5–1.0–2.0 mM S | 25 mM N | Increased antioxidant activity and decreased total phenolic concentration | [127] |
| *Fragaria ananassa* | 150–225–300–450–600 Kg ha$^{-1}$ N | 450–600 Kg ha$^{-1}$ N | Decreased solid soluble content and increased blemishes in fruits and acid content | [111] |
| *Ocimum basilicum* | N,P, K (1:1:1), 6 different levels (0, 25, 50, 75, 100 and 125 mg kg$^{-1}$ soil) | 50 mg Kg$^{-1}$ of soil | Increased total phenolic content | [128] |
| *Rubus ulmifolius* | 60–100 Kg ha$^{-1}$ N and 66.4–104 Kg ha$^{-1}$ K | 100 Kg ha$^{-1}$ N | Increased fructose, glucose, sucrose content, pH and secondary metabolites in fruit | [92] |

## 6. Soil P Supply: A Key Pre-Harvest Factor to Enhance the Quality, Condition and Antioxidants in Fruits

In addition to N, the P supply is a key factor for the fruit crop management because it affects the yield and modulates the soluble solids and secondary metabolites production, including ascorbic acids and phenols [148]. Moreover, the P nutrition can influence the postharvest physiology and the shelf-life of fresh fruits [108,149]. According to Afroz et al. [150], when appropriate P was added, the weight of strawberries increased (Table 2). Valentinuzzi et al. [151] found that soluble solids were lower in strawberry fruits with a P shortage, showing that a P deficiency could negatively affect the sugar biosynthesis. According to Cao et al. [152], the soluble solids in strawberries were positively related to the P content, as induced by the P supply. Noteworthy, Valentinuzzi et al. [151] concluded that the fruit firmness of strawberries grown under a P deficiency increased by 60% (Figure 2), which may be due to the higher Ca presented in the fruit tissue. It is also noteworthy that the P nutrition also induces significant changes in the bioactive compounds in fruits. In fact, the contents of vitamin C, malic acid, gallic acid, proline, lysine, sorbitol-6-phosphate, malate, and citrate were inhibited when P was deficient [150,151,153,154], whereas the anthocyanin contents were higher [151]. According to Matubara et al. [155], the P-deficient cells accumulated phenolic compounds, including caffeic acid and chlorogenic acid, and showed a PAL activity 25 times higher than those treated with normal levels of P. Ross et al. [156] also found, in sour cherry hybrids, that P increases were associated with a lower phenolic content. Increases in the phenol contents in low-P plants have also been reported in the apple [157] (Figure 2; Table 2). However, field experiments have shown that

the application of phosphate, a salt of phosphorous acid absorbed and transferred in a way similar to inorganic P, increased the anthocyanin content of strawberry fruits [158].

Kader [159] indicated that the P content could affect the postharvest respiration rate of fruit (Table 2). Indeed, Knowles et al. [149] observed that the respiration of low-P cucumber fruits was 21% higher than high-P fruits (Figure 2). According to Ge et al. [160], the trisodium phosphate (TSP) treatment could maintain the postharvest quality of the apple fruit by inhibiting the respiration intensity, delaying the weight loss, and inhibiting the decline of the flesh firmness, ascorbic acid, titratable acid, and SSC (Table 2). In agreement, Yogaratnam and Sharples [161] showed that the foliar applications of P during the growing season increased the tissue P concentration, thus reducing the incidence of storage disorders in apples. The membrane function in fruit tissue declines during ripening and senescence, which occur concomitantly with a loss of phospholipids or changes in the fatty acid composition [149,162]. In this way, Knowles et al. [149] reported that the low-P cucumber fruits had a lower concentration of phospholipids, a lower level of un-saturation in various pools of fatty acids, and a greater rate of electrolyte leakage, than that of high-P fruit. Interestingly, the fruit breakdown incidence and the P content have been reported negatively correlated in apple fruits [163]. According to Knowles et al. [149], the increase in susceptibility to the internal breakdown seems to result from the reduction of membrane stability caused by a decline in the phospholipid level. Olivos et al. [42] found that the nectarine fruit with low N, P, and K treatments had a higher flesh browning incidence expressed as the score and percentage (Table 2). The same study revealed that the fruit with deficient N, P, and K treatments had three times more fruit with flesh browning (50% to 60%) than fruit from the control treatment (20%) after storage for 11 d at 5 °C. In addition, the same authors showed that fruit with a low P treatment had a higher total phenolics and antioxidant activity than fruit from the other treatments [42]. Moreover, the PPO activity was significantly higher in fruit from the low P and K treatments, than in the other treatments, with browning potentially higher in the fruit from the low P treatment [42] (Table 2).

Finally, the browning incidence in fruits can be decreased through the correct balance between the N and P fertilization in the soil. Likewise, the relationship between both nutrient minerals is an important factor in the activity of the PPO enzymes and the concentration of the antioxidant compounds in fruits. However, it has been reported that foliar applications in apple plants with other nutrients, such as selenium (Se) have generated a decrease in the internal browning in fruits [38]. In the same way, applications of chitosan to sweet cherry fruits during the packaging process have shown a decrease in the PPO activity, fruit respiration and also a decrease in the degradation of the phenolic compounds during storage time, promoting a better postharvest condition. According to the reported data, it was possible to observe a decrease of approximately 30% in the parameters mentioned above for the treatments with the Se applications in pre-harvest and chitosan in post-harvest [38,164]. However, the fertilization program of an orchard is usually easier to correct and cheaper compared to other product applications, which are managed as an additional step to the conventional management of cherry orchards. Thus, authors have shown that with corrections of P and N in the soil and the foliar applications, it is possible to generate an increase in the antioxidant capacity and phenolic composition in the fruits of various fruit species, between 30 to 50% [42,89,95,150,151,153], promoting a greater postharvest fruit longevity.

**Table 2.** Studies about the impact of the phosphorus fertilization on antioxidants and the quality in fruits in different plant species.

| Species | Treatments | Phosphorus Treatment | Effects | References |
|---|---|---|---|---|
| *Prunus cerasus* × *Prunus canescens* | ammonium polyphosphate | 20 g P tree$^{-1}$ | Increased P concentrations in fruit, but had lower levels of phenolic compounds | [156] |
| *Fragaria ananassa* | P | 40–60 Kg ha$^{-1}$ | Higher effect on the fruit length, fruit diameter, and fruits per plants, and fruit weight | [150] |
| *Fragaria ananassa* | Control-Fe$^{-}$-P$^{-}$ | P$^{-}$ | Lower °Brix accumulation and higher phenolic compounds synthesis in fruits | [153] |
| | | | Higher malic acid, lower quinic acid, sorbitol-6-phosphate and quinic acid | |
| *Malus domestica* | Macronutrients 1/4-strength of MSmedium (1/4 MS), 1/2-strength (1/2 MS), full-strength (1 MS), two-fold (2 MS) and three-fold (3 MS) | 1/4-x MS and 1/2xMS | Higher concentration of anthocyanins in the shoots | [157] |
| *Fragaria ananassa* | 0–20–30–40-50% of P | 30% P | Higher anthocyanins concentration in the fruits | [158] |
| *Malus domestica* | 0–0.5 mg/mL TSP solution | 0.5 mg/TSP | Decreased weight loss, flesh firmness and respiration rate | [160] |
| *Prunus persica* | Control Full fertilized- Low N- Low P-Low K (hoagland's solution) | Low P | Decreased yield, fruit weight. Increased SSC, Antioxidant activity, Total Phenols, PPO and browning potential | [42] |
| *Cucumis sativus* L. | 90–360 mg P plant$^{-1}$ | 90 mg P plant$^{-1}$ | Increased fruit respiration rate at postharvest | [149] |
| Bramley's seedling | Control treatment- KH$_2$PO$_4$ sprays | KH$_2$PO$_4$ Sprays | Decreased core flush in fruits, senescent breakdown and superficial scald | [161] |

## 7. Conclusions

The sweet cherry production is a trend that will keep increasing in Southern Chile in future years, so it is necessary to understand every parameter to have a sustainable production, a higher fruit quality, and profitable yields. Internal browning is a critical quality parameter, which is more frequent where the cold storage time exceeds 45 days. Along the same line, many studies provide evidence for the negative effect of high nitrogen fertilization over the quality parameters and the increase of oxidative processes and the respiration rate of the fruit, triggering a decrease in the post-harvest potential. In the same way, a P deficiency demonstrates a positive effect on the antioxidant activity and the phenols composition, as well as an increase in the PPO activity and respiration rates, decreasing the storage potential. Therefore, this review points to the evidence for the value of a correct and balanced fertilization regime with different fruit tree species and the possible relationship of these studies to sweet cherry trees. This analysis could be important in the identification of the factor needed for avoiding the edaphoclimatic conditions of Southern Chile. Nonetheless, it is necessary to carry out more detailed field experiments to generate additional information to validate the antecedents presented in this review in the conditions of Southern Chile.

**Author Contributions:** C.P.-P. and A.R.-F. wrote the original manuscript; M.R.-D. and J.G.-V. reviewed and edited the manuscript. All authors have read and agreed to the published version of the manuscript.

**Funding:** The authors would like to thank CORFO-PTEC (Center of Southern Fruticulture; CODE: 16PTECFS-66647).

**Institutional Review Board Statement:** Not applicable.

**Informed Consent Statement:** Not applicable.

**Data Availability Statement:** Not applicable.

**Acknowledgments:** The authors would like to thank Jerry Cohen from the University of Minnesota for his critical revision and English editing.

**Conflicts of Interest:** The authors declare no conflict of interest.

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
