# Peer review of "The Potential Roles of the N and P Supplies on the Internal Browning Incidence in Sweet Cherries in the Southern Chile"

_horticulturae, doi:10.3390/horticulturae8121209_

Round 1
Reviewer 1 Report
1. The review is novel and valuable for the related research field but unfortunately it has errors in grammar and layout throughout the manuscript. Hence, it must be improved before it can be published.
2. A conclusive graph for presenting the key theme is suggested to enhance its readability and attractiveness.
3. L18: The Southern Chile - Southern Chile
4. L28: a highly phosphorus (P) fixation - a high phosphorus (P) fixation
5. L53: to protect orchards from rains - to protect orchards from rain
6. Tables 1 & 2: The layout and grammar should be improved. Scientific names should be in italics. Specie - Species; Effect - Effects
7. Add some photos of iconic plants, fruits, and most importantly the internal browning for sweet cherries in Southern Chile.
8. L321: southern Chile - Southern Chile
9. Table 2: Check this sentence "Higher concentration of anthocyanins in shhots"
10. Please rewrite it "Besides N, P supply is a key factor for fruit crop management due to it not only affects fruit yield but also modulates the production of soluble solids and secondary metabolites, including ascorbic acids and phenols."
Author Response
Temuco 21 November 2022
Editor of Horticulturae
Editorial Office of Horticulturae
Dear Editor of Horticultarae,
We have revised the manuscript entitled "The Potential Role of N and P Supply on the Internal Browning Incidence in Sweet Cherries in the Southern Chile" by C. Palacios- Peralta, Reyes-Díaz, M., J. Gonzalez-Villagra, and A. Ribera- Fonseca.
We would be very grateful if you would consider for publication in Horticulturae. The detailed answers to each point raised by the reviewers are below.
|
Reviewer #1 |
Responses |
|
The review is novel and valuable for the related research field but unfortunately it has errors in grammar and layout throughout the manuscript. Hence, it must be improved before it can be published.
|
Thank you very much for your opinion, we are very happy for it. We are agreed for the grammar of the manuscript and we have improved it. |
|
A conclusive graph for presenting the key theme is suggested to enhance its readability and attractiveness. |
Thank you very much for your suggestion. Now, we have included the graphical abstract to enhance the manuscript.
|
|
L18: The Southern Chile - Southern Chile |
Thank you for your correction. Done. |
|
L28: a highly phosphorus (P) fixation - a high phosphorus (P) fixation. |
Thank for you for your correction. Done. |
|
L53: to protect orchards from rains - to protect orchards from rain. |
Thank for you for your correction. Done. |
|
Tables 1 & 2: The layout and grammar should be improved. Scientific names should be in italics. Specie - Species; Effect - Effects |
Thank for you for your correction. Done. In addition, we have improved the grammar through the manuscript. |
|
Add some photos of iconic plants, fruits, and most importantly the internal browning for sweet cherries in Southern Chile |
Thank you for your suggestion. We are included the figure with photos. |
|
L321: southern Chile - Southern Chile |
Thank for you for your correction. Done. |
|
Table 2: Check this sentence "Higher concentration of anthocyanins in shhots" |
Thank for you for your correction. Done. |
|
Please rewrite it "Besides N, P supply is a key factor for fruit crop management due to it not only affects fruit yield but also modulates the production of soluble solids and secondary metabolites, including ascorbic acids and phenols. |
Thank for your suggestion, we have rewritten the sentence. |

Reviewer 2 Report
Major comments
1)-Title: As the work in devoted to the production of sweet cherry in Chile it is necessary to add ‘in the Southern Chili’ to the title
2)-Keywords: no words about cherry storage and browning
3)-Introduction: To attract the attention of readers it is highly desirable to indicate nutritional and health benefits of sweet cherry. In this respect at least two valuable references should be discussed and added:
a) D. S. Kelley, Y. Adkins, K. D. Laugero A Review of the Health Benefits of Cherries Nutrients. 2018,10(3): 368. doi: 10.3390/nu10030368
b) L M. McCune, Chieri Kubota,Nicole R. Stendell-Hollis,Cynthia A. ThomsonCherries and Health: A ReviewCritical Reviews in Food Science and Nutrition Vol 51, 2010 - Issue 1 https://doi.org/10.1080/10408390903001719
4)-Discussion
It is necessary to enumerate other methods for preventing fruit browning and compare their efficiency with N and P supply: chitosan, modified atmosphere of storage, aminobutyric acid. See:
‘K. Ruzickova, M Leitgeb The Role Of Browning Enzymes In Cherries/ Acta Innovations 2021 38(38):12-22DOI: 10.32933/ActaInnovations.38.2
-Indicate what methods of sweet cherry browning prevention are used in Chile
- as selenium supplementation is known to slow down senescence theoretically such an approach may also give a positive effect. See
Groth S, Budke C, Neugart S, Ackermann S, Kappenstein FS, Daum D, Rohn S. Influence of a Selenium Biofortification on Antioxidant Properties and Phenolic Compounds of Apples (Malus domestica). Antioxidants (Basel). 2020, 24;9(2):187. doi: 10.3390/antiox9020187.
5) Abstract
Taking into account the importance of comparison of different methods efficiency for prevention fruit browning I should recommend to make appropriate changes not only in the text but also in the Abstract and indicate the unusual approach of the authors about the importance of N and P supply optimization.
6)- References
Reference list should be carefully checked according to the authors guidelines:
Each reference should contain abbreviated journal title (see ref 55), volume number, pages (no pages in ref 121) and doi. -Don’t use Internet sites, like ‘Available online….’ (see ref 34, 39, 41 etc)
-there is no article title in Ref 23 and this reference contains the names of the authors but not their surnames
- no article title in ref 142
-don’t use capital letters in the articles titles (see ref 27, etc)
Minor comments
1)Line 116 - check style ‘due to it may …’
2) Line 204 change ‘NO3−, the highest dose (15 mol m-3)’ to ‘N03-, the highest dose (15 Mol m-3)
3)The same in Table 1 ‘NH4NO3’ and line 315 ‘mg kg-1 '
4)Al Latin names of plants should be written in Italics (Tables 1,e and References)
Line 258 ‘phosphite (Phi), a salt of phosphorous acid absorbed and transferred in a way similar to inorganic P’- a misprint: a salt of phosphoric acid is phosphate!!!!
Author Response
Temuco 21 November 2022
Editor of Horticulturae
Editorial Office of Horticulturae
Dear Editor of Horticultarae,
We have revised the manuscript entitled "The Potential Role of N and P Supply on the Internal Browning Incidence in Sweet Cherries in the Southern Chile" by C. Palacios- Peralta, Reyes-Díaz, M., J. Gonzalez-Villagra, and A. Ribera- Fonseca.
We would be very grateful if you would consider for publication in Horticulturae. The detailed answers to each point raised by the reviewers are below.
|
Reviewer #2 |
Responses |
|
Title: As the work in devoted to the production of sweet cherry in Chile it is necessary to add ‘in the Southern Chili’ to the title. |
Thank for your suggestion, we have included it on the title. |
|
Keywords: no words about cherry storage and browning. |
Thank you for the observation, we have included the word about these concepts. |
|
-Introduction: To attract the attention of readers it is highly desirable to indicate nutritional and health benefits of sweet cherry. In this respect at least two valuable references should be discussed and added:
a) D. S. Kelley, Y. Adkins, K. D. Laugero A Review of the Health Benefits of Cherries Nutrients. 2018,10(3): 368. doi:10.3390/nu10030368
b) L M. McCune, Chieri Kubota, Nicole R. Stendell-Hollis, Cynthia A. Thomson Cherries and Health: A Review Critical Reviews in Food Science and Nutrition Vol 51, 2010 - Issue 1 https://doi.org/10.1080/10408390903001719
|
Thank for your suggestion, we have included this point in the introduction. |
|
Discussion It is necessary to enumerate other methods for preventing fruit browning and compare their efficiency with N and P supply: chitosan, modified atmosphere of storage, aminobutyric acid. See: ‘K. Ruzickova, M Leitgeb The Role Of Browning Enzymes In Cherries/ Acta Innovations 2021 38(38):12-22DOI:10.32933/ActaInnovations.38.2
|
Thank you for your suggestion. We have included the paragraph that compare different methodologies to prevent browning of fruits. |
|
Indicate what methods of sweet cherry browning prevention are used in Chile. |
Thank you for your suggestion. We have included it. |
|
as selenium supplementation is known to slow down senescence theoretically such an approach may also give a positive effect. See Groth S, Budke C, Neugart S, Ackermann S, Kappenstein FS, Daum D, Rohn S. Influence of a Selenium Biofortification on Antioxidant Properties and Phenolic Compounds of Apples (Malus domestica). Antioxidants (Basel). 2020, 24;9(2):187. doi:10.3390/antiox9020187
|
Thank for the suggestion. We have included in the manuscript. |
|
Abstract Taking into account the importance of comparison of different methods efficiency for prevention fruit browning I should recommend to make appropriate changes not only in the text but also in the Abstract and indicate the unusual approach of the authors about the importance of N and P supply optimization. |
Thank you for the suggestion. We have included this issue in the abstract. |
|
References Reference list should be carefully checked according to the authors guidelines:
Each reference should contain abbreviated journal title (see ref 55), volume number, pages (no pages in ref 121) and doi. -Don’t use Internet sites, like ‘Available online….’ (see ref 34, 39, 41 etc).
|
Thank you for the corrections. We are checked all the references. |
|
there is no article title in Ref 23 and this reference contains the names of the authors but not their surnames
|
Thank you for the correction. Done. |
|
no article title in ref 142
|
Thank you for the correction. Done. |
|
don’t use capital letters in the articles titles (see ref 27, etc |
Thank you for the correction. Done. |
|
Minor comments Line 116 - check style ‘due to it may …’ |
Thank you for your correction. We have checked it. |
|
Line 204 change ‘NO3−, the highest dose (15 mol m-3)’ to ‘N0, the highest dose (15 Mol m-3) |
Thank you for your correction. We have changed. |
|
The same in Table 1 ‘NH4NO3’ and line 315 ‘mg kg-1 ' |
Thank you for the observation. We have changed. |
|
Al Latin names of plants should be written in Italics (Tables 1,e and References) |
Thank you for the comment. We are corrected through the manuscript. |
|
Line 258 ‘phosphite (Phi), a salt of phosphorous acid absorbed and transferred in a way similar to inorganic P’- a misprint: a salt of phosphoric acid is phosphate!!!! |
Thank you for the observation; it was a mistake, and we have corrected it. |
Sincerely Yours,
Dr. Alejandra Ribera-Fonseca and Dra. Marjorie Reyes-Díaz
Corresponding authors

Reviewer 3 Report
Dear Authors and Editors,
Based on the title, the topic of the review is the examination of a certain postharvest physiological problem, namely the internal browning incident of sweet cherry. The abstract reveals that this is mainly about the production in Chile. The manuscript contains a large number of references that are not strictly or directly related to the topic, and also the article itself does not deal closely with the topic indicated in the title.
The topic of the article is based on the assumption that the N and P content of the soil can influence the antioxidant content of sweet cherries, which reduces the probability of browning incidence. In order to support or refute this assumption, investigations would be needed with soil samples were taken from sweet cherry plantations, their N and P content, as well as the antioxidant content of the sweet cherry fruits were analyzed, and a correlation was sought between these and browning incidence.
The review article does not describe the results of a single experiment similar to the above procedure, rather it tries to prove the assumption with a kind of investigative method based on the results of experiments with a wide variety of fruit and vegetable plants. Another disadvantage of the manuscript is that it contains many unidentifiable references. Nevertheless, the manuscript also contains a lot of interesting and useful information.
I propose the rejection of the manuscript in its present form, and I encourage the authors to change the subject of the review, i.e. to choose a broader topic (e.g. “nutritional disorders and antioxidant content of plants” or “antioxidant content and resilience of plants”, etc.), and to write a thorough review that focus more on the given topic.
Author Response
Temuco 21 November 2022
Editor of Horticulturae
Editorial Office of Horticulturae
Dear Editor of Horticultarae,
We have revised the manuscript entitled "The Potential Role of N and P Supply on the Internal Browning Incidence in Sweet Cherries in the Southern Chile" by C. Palacios- Peralta, Reyes-Díaz, M., J. Gonzalez-Villagra, and A. Ribera- Fonseca.
We would be very grateful if you would consider for publication in Horticulturae. The detailed answers to each point raised by the reviewers are below.
|
Reviewer #3 |
Responses |
|
Based on the title, the topic of the review is the examination of a certain postharvest physiological problem, namely the internal browning incident of sweet cherry. The abstract reveals that this is mainly about the production in Chile. The manuscript contains a large number of references that are not strictly or directly related to the topic, and also the article itself does not deal closely with the topic indicated in the title. |
Thank you for your comments. We believe that the problem of browning incidence is a major problem worldwide for cherry-producing countries, especially in our country, which is one of the main producers. On the other hand, it is important to include other references in order to explain what has been observed and reported in Chile.
|
|
The topic of the article is based on the assumption that the N and P content of the soil can influence the antioxidant content of sweet cherries, which reduces the probability of browning incidence. In order to support or refute this assumption, investigations would be needed with soil samples were taken from sweet cherry plantations, their N and P content, as well as the antioxidant content of the sweet cherry fruits were analyzed, and a correlation was sought between these and browning incidence. |
Thank you for comment. We are agreed with your observation that the influence of N and P nutrition is an assumption; however, we are improved the review, including the comparison with other methodologies for decrease the browning incidence, discussed about the increase of antioxidant as a mechanism and also adding the word “potential” in the title. We believe that this topic is valuable and relevant for all cherry-producing countries, especially our country (the main producer). |
|
The review article does not describe the results of a single experiment similar to the above procedure, rather it tries to prove the assumption with a kind of investigative method based on the results of experiments with a wide variety of fruit and vegetable plants. Another disadvantage of the manuscript is that it contains many unidentifiable references. Nevertheless, the manuscript also contains a lot of interesting and useful information. |
Thank you for your comments. We have revised the references and as explaining above, we believe that the review has a lot useful and valuable information. In addition, with your and the other reviewers the manuscript was improved. |
|
I propose the rejection of the manuscript in its present form, and I encourage the authors to change the subject of the review, i.e. to choose a broader topic (e.g. “nutritional disorders and antioxidant content of plants” or “antioxidant content and resilience of plants”, etc.), and to write a thorough review that focus more on the given topic. |
Thank you for all comments from the reviewer, it certainly helped to improve the review; however, we think the topic is relevant, novel and valuable, as mentioned by the other two reviewers. It would be interesting, in the future, to be able to write a new review.
|
Sincerely Yours,
Dr. Alejandra Ribera-Fonseca and Dra. Marjorie Reyes-Díaz
Corresponding authors

Round 2
Reviewer 1 Report
The manuscript has been revised and I have no further questions.
Author Response
Dear Reviewer,
Thanks for your help in improving our manuscript.
Best wishes,
Dr. Marjorie Reyes and Dr. Alejandra Ribera
Corresponding authors

Reviewer 2 Report
There are only some minor comments:
1)Keywords: delete numbers
2)Line 345 ‘can decrease’ change to ‘can be decreased’
3)The Reference list should be revised: 1) use journals abbreviations, 2) use bold letters for the year of publication, 3) use Italics for the Journals’ titles and volumes
Author Response
Dear Reviewer,
Thanks for your help, valuable comments, and corrections, which helped to improve our manuscript.
Best wishes,
Dr. Marjorie Reyes and Dr. Alejandra Ribera
Corresponding authors

Reviewer 3 Report
Dear Authors and Editors,
I understand that sweet cherry browning incidence is a major problem. However, the vast majority of the references are not based on the results of experiments with sweet cherry, but with other fruits and vegetables. Only Sections 1 and 6 deal with sweet cherry production in Chile.
Section 2 “Physiological and Biochemical traits of Internal Browning in Fresh Fruits” explains the topic stated in the title very well, but it should be noted that most of the references are not based on sweet cherry research. Section 3 “Non-Enzymatic Antioxidant Compounds in Sweet Cherry Fruits.” has content corresponding to the title.
Section 4 “Nitrogen Excess could Reduce Quality, Condition and Antioxidant Levels in Fruits” is also an informative chapter in itself, but at the same time it is quite clear that the references contain only traces of information about sweet cherry, and Table 1 belonging to the chapter also shows facts about the most diverse plants apart from sweet cherry.
Section 5 describes results for strawberry, cucumber, apple, nectarine, and the related Table 2 does not include any reference to sweet cherry. Sweet cherry is mentioned only in one place in this section in connection with the Chitosan treatment, which is also not the topic of N and P supply in the title. At the end of the section, only references 44, 144, and tangentially 146 are valid (83 - no N or P supply, 89 - no antioxidant or phenolic composition, 83 - again, 143 – no antioxidant or phenolic composition), but these do not examine sweet cherry either.
Section 6 deals with the N and P content of Chilean soils and sweet cherry production, but does not provide any evidence that soil N and P supply is related to the sweet cherry browning incident.
The sections are put together like a patchwork and do not give a complete picture of what is stated in the title. By omitting some parts and expanding other parts, a more coherent text should be created, with a more suitable title.
Additional comments:
In line 56, the sentence does not continue after "however".
According to line 100-102, "fertilization doses of calcium (Ca), nitrogen (N), and phosphorus (P) can reduce IB in fresh fruits such as peaches [43–45]." –Reference 43 does not mention fertilization (it is about enzymatic browning of apple as a function of phenolic compounds) and reference 45 describes the effect of Ca spray on sweet cherry fruit quality and post-harvest storage, but only deals only with browning of stem and pedicel, and not fruit.
In line 361, the title of the next section is stuck.
Author Response
Dear Reviewer,
Thanks for your help, valuable comments, and corrections, which helped to improve our manuscript (please see the attachment with our responses).
Best wishes,
Dr. Marjorie Reyes and Dr. Alejandra Ribera
Corresponding authors

Round 3
Reviewer 3 Report
Dear Authors and Editors,
The authors made sufficient improvements, so I recommend to accept it.